



**Storming the news media: 5 years of reporting weather hazards and climate change**
**Chloe Brimicombe[1]**
**1 Department of Geography and Environmental Science, University of Reading, Reading,**
**RG6 6AB, UK.**
Correspondence to: Chloe Brimicombe c.r.brimicombe@pgr.reading.ac.uk
**Abstract:** Global heating has increased the risk of weather hazards in recent years.
Communication of weather hazard risk by the news media has importance. Newsworthiness
affects weather hazards reporting. Here, the methods used to adhere to the open science
principles of reproducibility and transparency. Methods used are advanced Google searches
of media articles and the emergency disaster database (EM-DAT) that consider the weather
hazards floods, heat waves, wildfires, storms and droughts. Storms have had a large number
of articles in the last five years. But, wildfires have a large number of articles per individual
occurrence. Science and media collaborations could address the bias and improve reporting.
**Plain Text Summary:**
Climate change is increasing the risk of weather hazards (i.e. Storms and Heatwaves). Using
open science methods it is shown that there is a bias in weather hazard reporting. Storms
have had a large number of articles in the last five years. But, wildfires have a large number
of articles per individual occurrence. Science and media collaborations could address the bias
and improve reporting.










## 1. Introduction

Weather hazards are having an increasing impact on our lives. The latest IPCC report demonstrates that storms, flooding, heat waves, wildfires and droughts have been increasing in intensity and frequency with climate change (IPCC,2021). The last 5 years has experienced a number of notable weather hazards, from the costly 2018 Pacific Typhoon season to the Pacific North West heat wave and European flooding in June 2021 and the Mediterranean heat wave and wildfire in August 2021 (Gao et al., 2020; Kreienkamp et al., 2021; Sjoukje Philip et al., 2021; Sullivan, 2021).

Communication of a risk does not always lead to the risk being understood (Porter and Evans, 2020), however the media is a key actor in communicating climate change and has a moral obligation to report all aspects of the climate emergency to highlight in this case the risk of extreme weather and what action is being taken (Boykoff and Yulsman, 2013; Kitzinger, 1999).In addition, it has previously been found that the media has often given more attention to outlier views on climate change, instead of the consensus view (Meah, 2019; Petersen et al., 2019).

Previous research demonstrates that the bias in reporting hazards and climate change leads to attention and material resource deficit, not fully recognising or addressing the risk (Brimicombe et al., 2021a; Howarth and Brooks, 2017).In comparison, it has been found that when visual hazards such as floods and storms (Wilby and Vaughan, 2011) are used to demonstrate climate change risk there is an improved understanding of climate risk, this is also known as objectifying climate change (Höijer, 2010).

In this study, open science principles (Armeni et al., 2021; Nosek et al., 2015) are adhered to whilst using simple advanced search tools provided by Google and the number of weather disasters as reported by the emergency database (EM-DAT) (CRED, 2020). This, allows for an examination of the English news media articles produced over the last 5 years to answer the key questions: Has there been an overall increase in articles in the last 5 years? What weather hazard had the most attention? And how many articles also discussed climate change?





## 2. Methods and Data

All the methods and data chosen by this study are in keeping with open data and open science. Open science is where the research results are reproducible and transparent (Armeni et al., 2021).

### 2.1 Advanced Google Search

An advanced Google search was carried out for the period 1st January 2017 to the 1st January 2022. The individual search selection was for all news articles in the period containing the keyword flood, heat wave, wildfire, storm and drought and then the search was carried out again this time including climate change as a keyword (cf. Brimicombe et al., 2021). Each hazard was evaluated separately and their results compared, with duplicated results not included.

Further, to counter any overestimates that might occur where articles are not discussing a weather hazard but are using the term to describe something else, the approach taken is to look at the first 100 articles headlines and remove articles not discussing a weather hazard. Examples included articles discussing 'Goal droughts', 'NFL Storm' and 'Glass Animals single Heatwave'. Then, this proportion of articles was removed from the overall total, giving a new overall count of articles. For example, for Storms in 2017, the initial search returned 6.31 million articles, but 21 out of the first 100 were not about the weather hazard. Therefore, 21% of the total articles were removed leaving 4.98 million articles.

Limitations of this method do remain it can still capture articles not explicitly about the weather hazard, however, this is limited by the proportional approach taken. In addition, it is only likely to capture the English news media and will give a slightly different number of articles between users. As such it is recommended that further in-depth research should be carried out looking at news media sentiment.



## 2.2 EM-DAT Hazard Reporting

To supplement the findings of the advanced google search, we use another source of data in keeping with open science, the emergency events database (EM-DAT). EM-DAT is the leading international disaster database, it contains details of over 22,000 mass disasters worldwide since 1900 and is compiled from a range of sources including UN agencies and Non-Governmental Organisations (NGOs) (CRED, 2020). This provides us with an overview of the number of weather hazards that have occurred every year for the last 5 years. This then allows us to assess on average how many articles have been written about each weather hazard. Table 1 shows a count of the weather hazards considered by this study included in EM-DAT (CRED, 2020).

*Table 1: Displaying the total number of disaster reported per weather hazard for the last 5 years as reported by EM-DAT (CRED, 2020).*

| *Weather Hazard* | *Number of Disasters reported in the last 5 years* |
|---|---|
| Drought | 64 |
| Flood | 865 |
| Heat wave | 38 |
| Storm | 557 |
| Wildfire | 66 |
| Total | 1590 |

Limitations of this method are that there are biases and under-reporting of hazards by this database(Brimicombe et al., 2021a; Gall et al., 2009). In addition, this database only includes hazards that are considered a disaster, where an agency declares a state of emergency, or where it is reported that over 100 people have been affected(CRED, 2020). However, it remains the most comprehensive source of reported weather hazards (Brimicombe et al., 2021a; Gall et al., 2009).





## 3. Results

### 3.1 Overall number of articles has increased

In total since 2017, over 142 million articles have been written by the English language news media about weather hazards. There has also been an increase in the number of English language news media articles for all weather hazards. Per year storms have the most articles, whilst heat waves have the least number of articles (Figure 1). The ranking of the total number of articles for each weather hazard type is storms, floods, wildfire, drought and heat wave. 28.1 million articles are about storms, whereas 169k articles are about heat waves in 2021 (Figure 1).

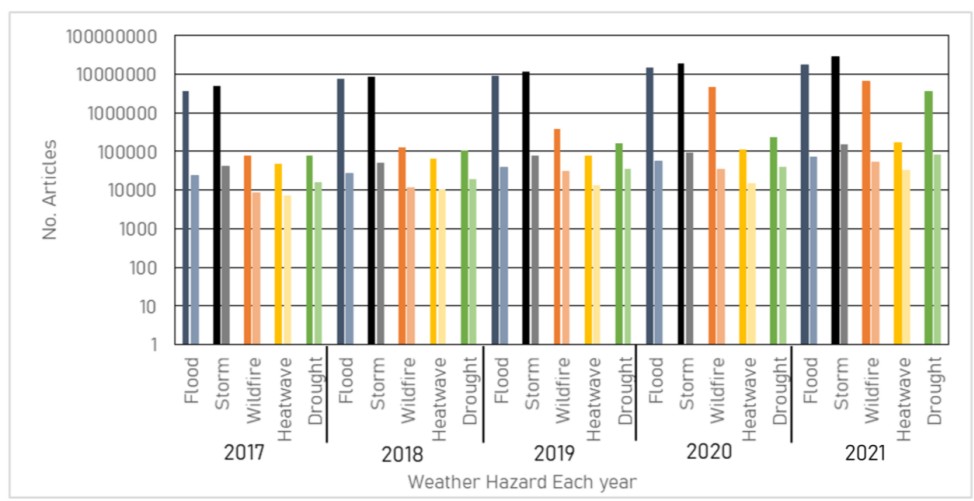

*Figure 1: Displays number of articles (on a logarithmic scale) per hazard per year for 2017 to 2021.The darker colour indicates overall article numbers whilst the lighter colour indicates only articles that contain the weather hazard and climate change as its subject.*

Fewer articles are about weather hazards and climate change at over 1.03 million. The number of articles about weather hazards and climate change has increased (Figure 1). Per year storms have the most articles, whilst heat waves have the least number of articles for weather hazards and climate change. In addition, the ranking of the number of articles for each weather hazard type is storms, floods, wildfire, drought and heat wave. In 2021, 149k articles include storms and climate change whereas 32k mention heat waves and climate change (Figure 1).





**3.2 Per hazard occurrence wildfire has the greatest number of articles**
The results in section 3.1 change when the number of articles is considered as a proportion
of the number of weather hazards reported in EM-DAT in table 1. Overall, on average for each
individual weather hazard, 89k articles were written, however, the picture for each hazard
varies widely. On average per wildfire, there have been 175k articles in the last 5 years (Figure
2). The weather hazard with on average the least number of articles per weather hazard
occurrence over the last 5 years are heat waves with 12k articles (Figure 2). The ranking of
the number of articles on average per weather hazard occurrence is wildfire, storm, drought,
flood and then heat wave.

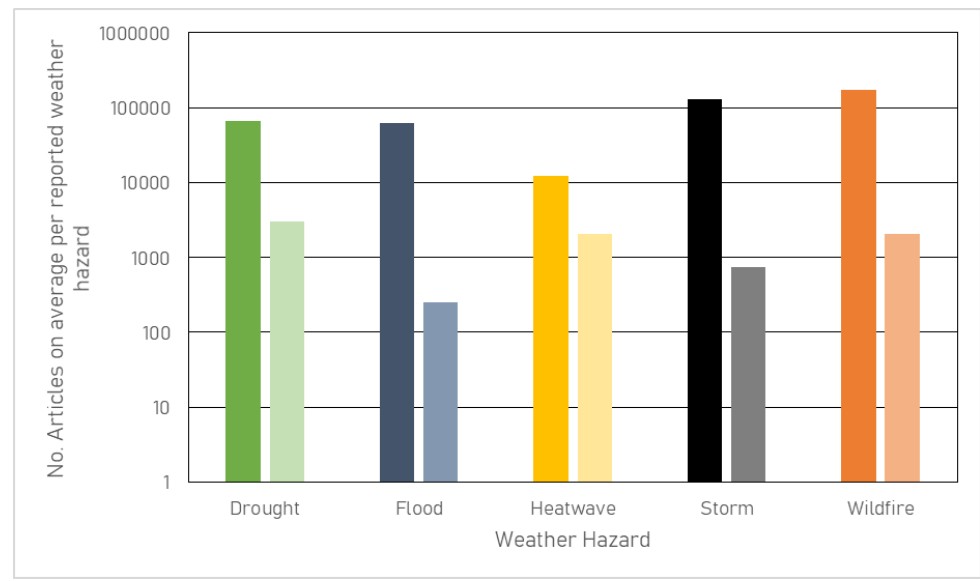


*Figure 2: displaying on average the total number of articles per reported weather hazard in*
*EM-DAT for the last 5 years (Logarithmic scale). Dark colours are all weather hazard articles,*
*whilst lighter colours are articles also including climate change.*







### 3.3 Individual droughts have the most articles discussing climate change

Overall, on average for each individual weather hazard, 650 articles were written that also consider climate change, however as with all weather hazard articles the picture for each hazard varies widely. On average per drought, there have been 3k articles in the last 5 years (Figure 2). The weather hazard with on average the least number of articles per weather hazard occurrence over the last 5 years are floods with 200 articles (Figure 2). The ranking of the number of articles that also consider climate change on average per weather hazard occurrence is drought, wildfire, heat wave, storm, floods.



## 172  4. Discussion

Heat waves have the least amount of news media articles. This should not be of surprise given
other research demonstrating the consistent underreporting of this weather hazard
(Harrington and Otto, 2020; Vogel et al., 2019). It however, may be of surprise given the
number of record-breaking heat waves during recent years such as the June 2021 Pacific
North-West heat wave which was found likely to of been impossible without Climate Change
(Sjoukje Philip et al., 2021).
How notable events or weather hazards get attention and are reported is subject to
'newsworthiness', which can also be known as the political economy between society and the
media (Boykoff and Yulsman, 2013; Kitzinger, 1999). This is made up of 4 main factors: *the*
*availability effect/heuristic which is if a hazard is presented as risk before it is more likely to*
*be remembered in this manner, stories from impacted groups, geographically bound and are*
*visually impactful* (Kitzinger, 1999; Tomlinson et al., 2011). The results of this study show that
the hazards that fit the criteria the most were storms which have the most articles by quantity
and wildfires that have the most articles per individual occurrence.
In addition, this study's results highlight a huge reporting bias in favour of storms and wildfire
in the news media. This attention bias in the overall number of reports has a material cost
where storms receive more research, funding and policy than other hazards (Brimicombe et
al., 2021b; Harrington and Otto, 2020; Howarth and Brooks, 2017; Vogel et al., 2019).
However, despite ranking second in terms of the overall number of articles, per individual
occurrence floods have the least number of articles. This is something that should be explored
further in a news media sentiment study.
In addition, the number of articles on average per individual weather hazard that also
considers climate change is not following the 'newsworthiness' criteria and therefore
drought, wildfire and heat waves have the most articles. Instead, the media can be suggested
to follow the science where it is seen these hazards are easier to attribute to climate change
than floods or storms (Ciavarella et al., 2020; Kreienkamp et al., 2021). Whilst the media does
have a moral obligation and plays a key role in communicating climate risk, how science, the
public and those in position of power communicates climate change has influence on what is



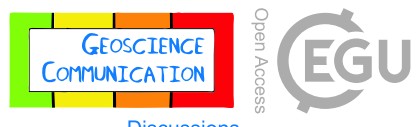

portrayed by the media (Boykoff and Yulsman, 2013; van der Hel et al., 2018; Howarth and
Anderson, 2019).
Therefore, it could be suggested that this reporting of climate change has come about by the
increasing collaboration between science and the media examples include Science Media
Centre, The Conversation and Voice of Young Science. This comes in spite of the discourse
around the role of science in both communication and policy spaces (Boykoff and Yulsman,
2013; Pielke, 2007).

**5. Conclusion**
The English News Media has a bias for weather hazards and climate change. Storm articles
have the largest total for the last five years, whilst wildfires have the most article per
individual hazard occurrence.
In comparison, storms have the most articles that also consider climate change. But, per
individual occurrence, drought articles is highest. Heat waves remain under-reported by the
English news media. Interestingly the number of flood articles is high. However, they are the
least reported per individual hazard. Exploring this along with the sentiment of news
reporting about weather hazards would be beneficial.
Weather hazards reporting remains subject to the newsworthiness factor and the political
economy of the media and society. The relationship between the media and science is
changing with climate change. Overall, the media should report the risk of climate change and
weather hazards. Science has a supporting role to play through collaborations with the media.
**Disclosure Statement:**
*The authors report there are no competing interests to declare.*
**Data availability:**
All data is available via advance Google searches and the EM-DAT database.






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
