# Peer review of "Is there a climate change reporting bias? A case study of English language news articles,"

_Geoscience Communication, 2022_

## Editor Decision (ED1)

**Gc-2022-8 Final decision**

Thanks to both the reviewers for addressing the vulnerable points of this paper. In particular, I would like to emphasize what suggested reviewer #1 when writing:

 "the results section is very hard to read. I had to read many sentences several times to fully understand what was being communicated. My advice is to divide the results section into three parts: (1) have the number of weather hazards news article increased since 2017?; (2) which weather hazards receive the most attention in news articles?; and (3) how often is climate change discussed in these news articles in relation to weather hazards?. Each section could be discussed in 3 or 4 sentences, giving more space for the discussion."

I would like also to stress that it is important to reorganize the figures and tables in order to help the reader to understand at a glance which are the results of the research. To this respect what suggest rev.#1 for Table1 is of pivotal importance (having also the data for the weather hazards per year would help the reader to compare the frequency of the events with the frequency of the reporting). The author can also consider to add a final table to summarize all the data to motivate the bias found in the media reporting extreme weather events once clarified why it is important to address it.

Also adding a paragraph on the approximate damage caused by different weather hazards in the last five years would add value to the article as suggested by rev#2, since, as we know, damage amount is what very often makes an event newsworthy.

Being confident that Brimicombe will fulfil all the reviewers 'requests, I will be happy to read a more organized version of this paper before accepting for publication.

---

## Editor Decision (ED2)

[revised manuscript text omitted]

This paragraph still remain confusing…Please explain how did you obtain these numbers, and if this is the case make again reference to table 1

[Figure]

*Figure 2: The average number of articles per individual hazard category for the last 5 years. Dark colour is total number of articles and light colour is articles including climate change.*

**3.3 how often is climate change discussed in these news articles in relation to weather hazards?**

Overall, on average for each individual weather hazard, 650 articles were written that also consider climate change.  The hazard with the most articles written is drought, on average per drought, there have been 3k articles in the last 5 years (Figure 2). The weather hazard with on average the least number of articles per weather hazard occurrence over the last 5 years are floods with 200 articles (Figure 2).

Again, please explain how did you obtain these numbers, and if this is the case make again reference to table 1

[revised manuscript text omitted]

---

## Author Response (AR3)

| | |
|---|---|
| This is an interesting topic and deserves critical attention. Yet the present manuscript muddles several things together, which makes it difficult to discern if the research aims have been met, and what the overall take-home messages are. Below I have made some suggestions to improve the readability of the piece. | Thank you for all your comments. I hope by addressing them the manuscript will be improved. |
| First, I think the title needs to be revised. Unless you have read the article it's unclear what the 'storming the news media' means, and even after reading the article I'm not sure this is the central finding. A title that is clearer (and more descriptive) would be preferable. For example, "Is there a climate change reporting bias? A case study of English language news articles, 2017-2022". | I agree and have change the title. |
| Second, the results section is very hard to read. I had to read many sentences several times to fully understand what was being communicated. My advice is to divide the results section into three parts: (1) have the number of weather hazards news article increased since 2017?; (2) which weather hazards receive the most attention in news articles?; and (3) how often is climate change discussed in these news articles in relation to weather hazards?. Each section could be discussed in 3 or 4 sentences, giving more space for the discussion. | Thank you for your comment. I have restructured the results section as you suggest. |
| Third, the conclusion should be rewritten. Rather than repeat the findings, tell the reader what the findings mean and why they matter. Why does it matter if this a bias in reporting extreme weather events? Is it because people may be left unprepared for one risk over another? Money may be invested in one problem compared to another? Or because invisible risks continue to persist until they reach a critical tipping point? Without this, does it matter if floods are reported more | Thank you for your comment. I have included this in the updated conclusion and discussion section. |

| | |
|---|---|
| than heatwaves? And what should be done about this? | |
| Lastly, there are some questionable calculations throughout the manuscript, where the aggregate findings are correlated with other data points. I'm mindful that, on face value, the calculations may be a little misleading, and worst, meaningless (see comments below). It's a pity that no information is provided on where the weather hazards occurred, or where the news articles focus their attention. Do they, for example, only report on storms in Europe? Overall, I think the paper is trying to do too much. My advice is to strip back much of the results and wider findings and concentrate on what was done, what is reliable, and what this tells us. Also, in trying an experimental analysis, what can other researchers learn from this experience? | Thank you, my previous research (Brimicombe et al., 2021 *Earths Future*) demonstrates that EM-DAT and often reporting is European centred. I however, agree with you that this can be confusing and misleading which was not my intension. I have sort to clarify this given R2s suggestions. |
| • Delete line 7. It's unclear and adds little. | Thank you I will delete it. |
| • Rephrase lines 8 and 9 to: "How weather hazards are communicated by the media is important. Which risks are understood, prioritised, and acted upon, can be influenced by the level of attention they receive. In this paper,…". | Thank you I will change this. |
| • Rephrase line 12 to: "hazards floods, heat waves, wildfires, storms and droughts from 2017-2022". | Thank you I will change this. |
| • Rephrase lines 12, 13 and 14 to: "Storms are more likely to be reported than any other climate risk. But wildfires generate more news articles per event. Bias in reporting needs to be addressed. | Thank you I will change this. |
| • Lines 30, 31, 32, 33, 34, and 35 change to "The IPCC's AR6 report… have increased in intensity and | Thank you I will change this. |

| | |
|---|---|
| frequency (IPCC 2021). Since 2017, there have been a number of notable weather events: Pacific Typhoon season 2018, European floods in 2021, Mediterranean heatwave and wildfires in 2021". | |
| • Line 39 and 40 change to: "to highlight the risk of extreme weather and what action is needed". | Thank you I will change this. |
| • Line 41 delete "previously". | Thank you I will delete this. |
| • Line 41 change to "found that the media often gives". | Thank you I will change this. |
| • Line 42 change: "outlier" to "sensationalist" | Thank you I will change this. |
| • Line 44 delete "previous". | Thank you I will delete this. |
| • Line 48 delete "this is". | Thank you I will delete this. |
| • Line 50-54 change to: "Reported here for the first time, this study uses open science principles (Armeni et al. 2021; Nosek et al. 2015) alongside the advanced search tools provided by Google, and the emergency database (EM-DAT) (CRED 2020), to examine how weather hazards are mentioned in news articles, from 2017-22. The aim is to understand: (1) has the number of articles focused on weather hazards increased since 2017; (2) which weather hazards receive the most attention; and (3) how often is climate change discussed in relation to those weather hazards". | Thank you I will change this. |
| • Line 60: Not sure this is the correct definition of 'open science'. | I agree – this is a description more in keeping with open data than open science which are related I will make this clearer. |
| • Line 64 change to: "The search involved two stages: first, a search for all news articles in the period containing keywords – flood, heat wave, wildfire, storm and drought, was conducted, and second, this search criterion was repeated with the keywords – climate change". | Thank you I will change this. |
| • Line 69 delete "Further". | Thank you I will delete this. |

| | |
|---|---|
| Lines 73-76, I'm not sure the logic for removing 21% of the articles makes sense. This calculation assumes that there is an even distribution of relevant and irrelevant news articles not only in the sample chosen, but also the rest of the articles collected. A clearer explanation is needed here. | The reasoning for this is because when writing a shorter manuscript using a similar method https://gc.copernicus.org/preprints/gc-2021-27/ the editor found that especially for Storm many of the articles were not about the weather hazard. I added this step to the method to arbitrarily simply try and address articles that may not be about the hazard and to avoid overestimation of number of articles. I will make this clearer. |
| Line 77 is unclear. Please rephrase. | Thank you I will incorporate this with restructuring lines 73 to 77. |
| How do you know the search results returned from Google are news-articles? What process was involved in determining that the articles came from news outlets and were written by journalists? | Thank you, all articles are filtered using the google search selection of news. I will add this to the method. |
| Line 87 – who is the 'we' in this sentence? Or 'us' in line 91 and 93? | Thank you, I will change this to either third person or singular. |
| Could Table 1 be reformatted so that it includes the data for the weather hazards per year? For example, a column for 2017, 2018, 2019, 2020, and 2021. Then the reader can learn the frequency of the events – before understanding the frequency of the reporting. That is, there were 10 droughts in 2017. | Thank you, I will look to restructure this and follow your suggestions from a previous comments of how this could be misleading currently. |
| Line 99: what are the biases? | Biases exist in what disasters are recorded in the EM-DAT database (Brimicombe et al., 2021a; Gall et al., 2009) I will make this clearer. |
| Line 111 is imprecise. Do you think the number of articles has increased year-on-year? That the number in 2021 is more than in 2017? Or something else? The current phrasing suggests a comparative dimension, but what is being compared? No data is provided to pre-2017. | Thank you I will make this clearer as a summary of the total number of articles. I mean over the 5 years here the aggregated total (the sum of the number of articles each year since 2017) is over 142 million articles. |
| Line 112: "More articles are written about storms each year compared to other weather hazards. Whereas the fewest number of articles were written about heatwaves each year". | Thank you, I will make this clearer. |
| Line 115: write the full figure 169,000. | Thank you, I will change this. |
| Figure 1 is visually deceptive. Could a different style be used to illustrate the | Thank you. I will look to see if there is another way to visualise this. |

| | |
|---|---|
| vast differences in reporting? Perhaps a treemap? Or a visual where the risks are represented by their size? | |
| Line 122 is confusing to read. Are you saying that out of the 142 million articles, only 1.03 million discuss climate change and the weather hazard? If that is the case, you really should showcase it. "Of interest, only 0.7% of all news articles mentioned climate change and the weather hazard together. This type of reporting has increased year-on-on-year, however". | Thank you. Yes that is what I was trying to highlight I will rephrase. |
| Line 123-128 do not add much. Do you need to explain the rank order in relation to climate change? Is there a difference? If not, delete this. | Thank you. I will incorporate the most useful information with the previous paragraph. |
| Lines 130-137, I am not convinced about the logic here. The disproportionate number of articles focused on 'storms' will make any correlations to events potentially misleading. Moreover, one 'storm' might generate 10x more articles than another, but this nuance is lost by aggregating the totals. I do not think this analysis adds anything, and instead it creates confusion. Indeed, this analysis is not needed to answer the paper's research aims. | Thank you. I will look to restructure this as I stated above.

The reason I aggregated the totals is because of the underreporting of some hazards in comparison to others across science reports, the media and EM-DAT as discussed in the method. I will look to make sure this is not misleading.

When looking at the average of number articles per total hazards across the 5 years we are able to reveal a trend that warm hazards are more readily attributed to climate change than wet hazards, which we don't see in Figure 1. |
| Lines 147-154 could be condensed in two sentences that explain that drought articles were more likely to include mentions to climate change, and floods the least. | Thank you I will look to restructure. |
| Line 177 change to: "which was attributed to climate change". | Thank you I will change this. |
| Line 194 delete "in addition". | Thank you I will delete this. |
| Why is that the media are happier to attribute climate change to hazards associated with warming (heatwaves, droughts) and less willing to do so for hazards associated with wetness (floods, storms)? | Thank you I state in Line 196: Instead, the media can be suggested to follow the science where it is seen these hazards are easier to attribute to climate change than floods or storms (Ciavarella et al., 2020; Kreienkamp et al., 2021).
But will look to make this clearer. |

| | |
|---|---|
| • Line 210 change to: "There is a bias in terms of which weather hazards English language news media report on, and a bias in terms of which weather hazards are linked to climate change". | Thank you I will change this. |
| • A lot of space is dedicated to explaining if journalists link extreme weather events and climate change. But why is this important? Why does it matter if certain weather events are attributed more often to climate change than others? | Thank you, Line 188: This attention bias in the overall number of reports has a material cost where storms receive more research, funding and policy than other hazards.
But as above I will make this clearer in the discussion and conclusion. |
| • How were articles treated that focused on primarily on one weather event but mentioned others too? | Thank you. I will add this to the method section. They are counted twice. |

Reviewer 2:

| | |
|---|---|
| This is an interesting topic for me. The manuscript is well written and compiled a good amount of data. | Thank you. |
| Major comment:
- The Author could also add the approximate
damage caused by different weather hazards
in the last five years. This will certainly increase the impact of the article and also would correlate with how 'number' alone is not the key point to be reported in the media but corresponding damage amount also can play the role. If possible, I suggest to the author add a section on it. | I have now included this in section 3.2 thank you. |
| The conclusion section should be rewritten, not repeating the findings, but looking what is the implication of the findings and so on. | I have changed this. |
| In the conclusion author also can discuss. Is there a geographical bias for these kinds of hazards? | Thank you, I've added it into the discussion. |
| Re-write the caption for Figure 2. | I agree and have changed. |
| Line 30: Define IPCC | It has been defined. |
| Line 41: delete 'previously' | Thank you, this has been removed. |
| Line 48: delete 'this is' | Thank you this has been removed. |
| Line 50: add a sentence of open science principles. | I agree, and have expanded this. |
| Line 63-68: rephrase the paragraph.
same
for Line 77 | Thank you, these have been rephrased. |

Editor:

| Thanks to both the reviewers for addressing the vulnerable points of this paper. In particular, I would like to emphasize what suggested reviewer #1 when writing: "the results section is very hard to read. I had to read many sentences several times to fully understand what was being communicated. My advice is to divide the results section into three parts: (1) have the number of weather hazards news article increased since 2017?; (2) which weather hazards receive the most attention in news articles?; and (3) how often is climate change discussed in these news articles in relation to weather hazards?. Each section could be discussed in 3 or 4 sentences, giving more space for the discussion." | I also thank the reviewers for their constructive comments. I have taken them onboard.

I have restructured the results like this. |
|---|---|
| I would like also to stress that it is important to reorganize the figures and tables in order to help the reader to understand at a glance which are the results of the research. To this respect what suggest rev.#1 for Table1 is of pivotal importance (having also the data for the weather hazards per year would help the reader to compare the frequency of the events with the frequency of the reporting). The author can also consider to add a final table to summarize all the data to motivate the bias found in the media reporting extreme weather events once clarified why it is important to address it. | I agree and have restructured to make this clearer and focused on the importance in the discussion. |
| Also adding a paragraph on the approximate damage caused by different weather hazards in the last five years would add value to the article as suggested by rev#2, since, as we know, damage amount is what very often makes an event newsworthy. | This has been added to a results section. And was a really good suggestion by R2. |
| Being confident that Brimicombe will fulfil all the reviewers 'requests, I will be happy to read a more organized version of this paper before accepting for publication. | Thank you for your confidence. I hope my revisions will allow for publication, I think this an important area that this research will expand the discussion in, to reduce reporting bias. |

**Many thanks for refining your work based on the reviewers (and editor) comments. However there are still a few comments made by the handling editor that need to be addressed before acceptance for publication. Please review and address these comments which were sent to you on August 12, 2022. The concerns raised on page 7 and 8 (result sections 3.2 and 3.3) are particularly important as they relate to the number of articles you have reported.**

*Thank you. I've added and made changes based on the handling editor technical notes, I've added an equation to the first time a proportional result is presented in each result section and updated the figure caption to the same effect. For example Line 136 "Overall, on average for each individual weather hazard (Total number of articles for all hazards in Figure 2/Total number of reported hazards in figure 1)."*

**A few additional (but minor) edits:**

**1. Please place a comma every third digit to the left of the decimal point when reporting large numbers to help the reader. I see you have done this mainly in the manuscript text and not in the figures.**

*I've now done this across the text and figures.*

**2. Is it possible to present the data in Table 1 and 2 similarly to what you have done in Figure 1 and 2? It is not easy to read and immediately see patterns in numbers in Table 1 and 2, hence the suggestion to present them differently.**

*I agree and I've formatted Table 1 and 2 to be figures as suggested, I hope this aids with presenting the patterns.*

**3. Use the word "heat wave" consistently throughout the manuscript. There are places where you write it as "heatwave".**

*Thank you, I've now changed this throughout the body of the manuscript.*

**4. Why did you select the google search engine for this analysis (as opposed to other search engines)? How does your selection of google search engine impact your results? Perhaps explain this in your method section so that it is clear.**

*This is a good reflection, I've now added to the method section to account for this:" Google was chosen as it has the most comprehensive results in comparison to other search engines (i.e. Bing) and tools that assisted with advanced search. " Line 66*

**5. Your analysis included articles with the word "climate change". How about articles that use different terms such as "climate crises, global warming or climate warming" to refer to "climate change"? Did you take these into accounts, and if not, how would omitting these terms impact your analysis?**

*Another good reflection the method used here uses the term that when I evaluated which key words to use returned the most results and I've now added this to the method section." Each*

*term was assessed to consider whether it captured the most articles, for example using heat wave not heatwave and climate change not climate crisis or global warming." Line 71*